# Regulation of life span by the gut microbiota in the short-lived African turquoise killifish

Patrick Smith[1†], David Willemsen[1†], Miriam Popkes[1†], Franziska Metge[1], Edson Gandiwa[2], Martin Reichard[3], Dario Riccardo Valenzano[1,4*]

[1]Max Planck Institute for Biology of Ageing, Cologne, Germany; [2]Chinhoyi University of Technology, Chinhoyi, Zimbabwe; [3]Institute of Vertebrate Biology, Czech Academy of Sciences, Brno, Czech Republic; [4]CECAD, University of Cologne, Cologne, Germany

**Abstract** Gut bacteria occupy the interface between the organism and the external environment, contributing to homeostasis and disease. Yet, the causal role of the gut microbiota during host aging is largely unexplored. Here, using the African turquoise killifish (*Nothobranchius furzeri*), a naturally short-lived vertebrate, we show that the gut microbiota plays a key role in modulating vertebrate life span. Recolonizing the gut of middle-age individuals with bacteria from young donors resulted in life span extension and delayed behavioral decline. This intervention prevented the decrease in microbial diversity associated with host aging and maintained a young-like gut bacterial community, characterized by overrepresentation of the key genera *Exiguobacterium*, *Planococcus*, *Propionigenium* and *Psychrobacter*. Our findings demonstrate that the natural microbial gut community of young individuals can causally induce long-lasting beneficial systemic effects that lead to life span extension in a vertebrate model.

*For correspondence:
dvalenzano@age.mpg.de

†These authors contributed equally to this work

Competing interests: The authors declare that no competing interests exist.

## Introduction

Life expectancy of different species in nature is regulated by a complex combination of genetic and non-genetic factors. Genetic manipulations in model organisms have revealed key conserved molecular pathways, including the insulin-IGF1 and the mTOR pathways, which regulate aging and life span across several species, spanning from yeast to mammals (*Kapahi et al., 2010*; *Kenyon et al., 1993*; *Lapierre and Hansen, 2012*). Environmental interventions such as temperature and dietary manipulations have also been importantly associated with life span modulation in several species. Among these, lower temperatures (*Conti et al., 2006*; *Miquel et al., 1976*; *Valenzano et al., 2006a*; *Van Voorhies and Ward, 1999*) and reduced nutrient intake (*Fontana et al., 2010*; *Mair and Dillin, 2008*) are key environmental factors that have been associated with prolonged life span.

Complex microbial communities covering external surfaces live at the interface between organisms and the external environment – from roots and leaves in plants, to skin, mucosal surfaces and gut in animals. These microbial communities participate in a wide range of key biological processes, including nutrient absorption (*Semova et al., 2012*), development (*Sommer and Bäckhed, 2013*), metabolism (*Nicholson et al., 2012*), immune modulation (*Geva-Zatorsky et al., 2017*), defense against pathogens (*Kamada et al., 2013*; *Schuijt et al., 2016*) and disease (*Sampson et al., 2016*).

Individual gut microbiota (GM) composition changes dramatically in various diseases (*Baumgart and Carding, 2007*; *Garrett, 2015*; *Sokol et al., 2008*) and during aging in flies, mice and humans (*Clark et al., 2015*; *Langille et al., 2014*; *O'Toole and Jeffery, 2015*). Healthy GM is

**eLife digest** Our bodies are home to lots of microorganisms, many of which are found throughout the gut. Gut microbes play important roles in human health, where they cooperate with our own cells to develop the immune system, synthesize essential vitamins, and help to absorb nutrients. When the cooperation between our own cells and the gut microbes fails, the microbial community within the gut can become a source of infection, sometimes leading to life-threatening diseases.

Healthy individuals typically have many different types gut microbes, whereas people with poor health, or older individuals, will often have less diverse and a higher percentage of disease-causing microbes. For example, African turquoise killifish only live a few months, during which the composition of their gut microbes undergoes dramatic changes. While young fish harbor highly diverse microbial communities, older fish have less diverse communities and more microbes associated with disease. Until now, it was not known whether manipulating the gut composition could affect the aging process.

By using the killifish as a model for their study, Smith et al. revealed that gut microbes affect how the fish survived and aged. When the guts of middle-aged fish were colonized with microbes transferred from younger fish, the older fish lived longer and were more active later in life. These fish also maintained a more diverse microbial community throughout their adulthood and shared key microbes with young fish – possibly associated with the improved health benefits. These results suggest that controlling the composition of the gut microbes can improve health and increase life span.

The model system used in this study could provide new ways to manipulate the gut microbial community and gain key insights into how the gut microbes affect aging. Manipulating gut microbes to resemble a community found in young individuals could be a strategy to delay the onset of age-related diseases.

typically characterized by large bacterial taxonomic diversity, whereas frailty is associated with loss of diversity and expansion of more pathogenic bacterial species (*Claesson et al., 2012*). Following antibiotic treatment, pathogenic bacterial species, such as *Clostridium difficile* and *Enterococcus faecalis*, can restructure the human GM and cause severe chronic conditions that pose a major threat for public health (*Bäckhed et al., 2012*; *Cox and Blaser, 2015*). Studies across different human age cohorts have shown that large changes in the abundance of subdominant bacterial taxa in the gut are a hallmark of aging; moreover, exceptionally long-lived individuals – including supercentenarians – are characterized by the persistence of bacterial taxa associated with a more healthy status (*Biagi et al., 2016*). While diversity-associated microbial taxa often decline during age, specific bacterial taxa, such as Clostridiales, are associated with malnutrition and increased frailty (*O'Toole and Jeffery, 2015*). In flies, reducing GM dysbiosis by improving immune homeostasis promotes longer life span (*Guo et al., 2014*; *Li et al., 2016*). Manipulating the GM towards a healthy state has the therapeutic potential to improve health in specific diseases (*Dodin and Katz, 2014*; *Kunde et al., 2013*). However, due to the lack of suitable short-lived vertebrate experimental models, it is not known whether age-associated gut microbial community changes causally affect the aging process and whether resetting a young GM in middle-age individuals can improve long-term health and affect individual life span in normal aging individuals.

In this study, we develop the turquoise killifish (*Nothobranchius furzeri*), a naturally short-lived vertebrate species with a life span of a few months in captivity (*Valenzano et al., 2015*), as a new model organism to study aging in the host gut and microbiota. We show that turquoise killifish (TK) have a complex GM (both in the wild and in captivity), similar in taxonomic diversity to that of mammals. We also show that during aging the overall microbial diversity in the TK gut decreases, with increased over-representation of pathogenic Proteobacteria. By acutely recolonizing middle-age individuals with GM from young donors, we developed an intervention that enabled fish to live significantly longer, remain more active at old age, and maintain a highly diverse GM. Transcriptome analysis additionally revealed that gut aging is associated with increased inflammation and reduced

proliferation. Here we provide the first evidence that acute gut microbiota transfer in the context of normal aging can significantly prolong life span in a vertebrate, becoming a novel candidate life span enhancing intervention. Our study also promotes the turquoise killifish as a highly suitable vertebrate model system to study the crosstalk between intestine and gut microbiota during host aging.

## Results

### The TK is a short-lived vertebrate with a complex GM

The TK is a naturally short-lived vertebrate, whose genome sequence has become available (*Reichwald et al., 2015*; *Valenzano et al., 2015*) and that is amenable to genetic manipulations via transgenesis or genome editing (*Harel et al., 2015*; *Valenzano et al., 2011*). Remarkably, this species is characterized by a broad spectrum of aging phenotypes, including cancer, neurodegeneration, and behavioral decline (*Harel and Brunet, 2015*; *Kim et al., 2016*). TK are adapted to living in conditions of intermittent availability of water and to surviving during brief rainy seasons and long dry seasons (*Cellerino et al., 2016*). In captivity, it lives between four to eight months, depending on the strain (*Valenzano et al., 2015*). However, nothing is known about its associated commensal microbes and whether its gut microbial complexity and taxonomic richness matches that of short-lived invertebrate model organisms, such as worms and flies, or that of longer-lived vertebrate model organisms, such as mice and zebrafish. To determine the TK's gut microbial composition, we sequenced the hyper-variable V3/V4 regions of the 16S rRNA gene amplicon from intestines of captive TK (n = 11, Materials and methods). We found that TK are characterized by a gut microbial taxonomic diversity similar to other, longer-lived vertebrates, including zebrafish, mice and humans (*Kostic et al., 2013*; *Qin et al., 2010*; *Stephens et al., 2016*) (*Figure 1A*, *Figure 1—source data 1* and *2*). This bacterial taxonomic diversity is an order of magnitude higher than the gut microbial diversity present in invertebrate model organisms such as worms (*Cabreiro and Gems, 2013*) and flies (*Buchon et al., 2013*) in the laboratory (*Figure 1A*). Remarkably, the four most abundant bacterial phyla present in the TK's gut, i.e. Proteobacteria, Firmicutes, Actinobacteria and Bacteroidetes, are also the four most abundant human gut bacterial divisions (*Zoetendal et al., 2006*) (*Figure 1B* and *Figure 1—source data 1* and *2*), although in different proportions (*Figure 1A*). The unique combination of a complex gut microbial composition, similar to that of other vertebrate aging model organisms, and its naturally short life span, combined with a wide spectrum of aging phenotypes, makes the turquoise killifish an ideal system to study the role of the gut microbiota during vertebrate aging.

### Wild and captive TK populations share a core GM

To assess whether the GM of captive TK was representative of the microbial communities associated with wild populations, we sequenced 16S rRNA gene amplicons from individual fish that we collected from different localities in the natural habitat of this species, ranging from the Gonarezhou National Park in Zimbabwe to the Gaza region in Mozambique (*Figure 2A*, *Figure 2—figure supplement 1B*, *Figure 2—source data 1*, Materials and methods). Sequencing these wild populations confirmed that Proteobacteria was the dominant phylum also in most wild populations, similar to laboratory fish (*Figure 2B*, *Figure 2—figure supplement 1C*). Individuals from all wild populations had a more diverse GM than laboratory fish (*Figure 2C*, Shannon alpha diversity, Dunn Kruskal-Wallis test, BH-adjusted p values < 0.05). While standard frequency-based diversity measures of observed bacterial taxonomic units (OTUs) (Simpson's and phylogenetic alpha diversity across the whole tree, observed OTUs, *Figure 2—figure supplement 1A*) were higher in wild populations, Chao1 alpha diversity, which gives more weight to rare bacterial OTUs, was higher in laboratory fish (*Figure 2—figure supplement 1A*). These results possibly reflect the fact that laboratory fish are dominated by few, high-abundance OTUs, hence resulting in having more 'rare' bacterial taxonomic units compared to wild populations. Differences in OTU abundance between laboratory-raised and wild fish might reflect ecological differences between the standardized laboratory conditions and the more heterogeneous wild environment, characterized by fluctuations in temperature, nutrients as well as other biotic and abiotic factors (*Blažek et al., 2017*). To test whether differences in ecology across distinct wild fish populations influenced bacterial diversity levels, we ran a regression model between

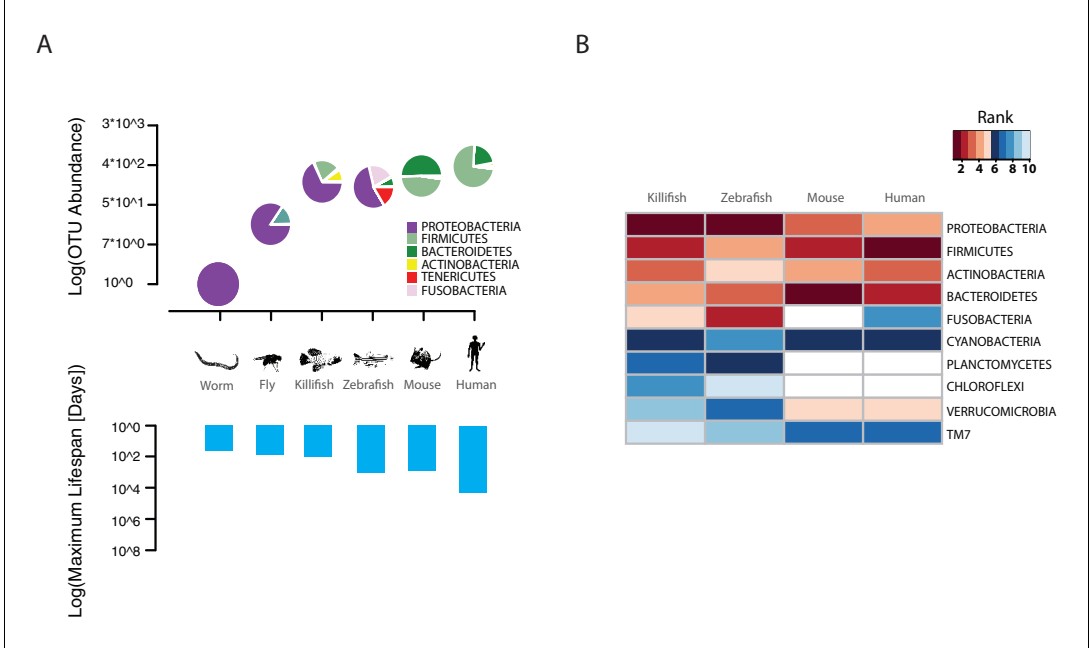

**Figure 1.** TK as a model to study aging of the GM. (**A**) Above: relative bacterial OTU composition (pie charts) and diversity (y-axis of upper plot) across different aging model organisms (*Figure 1—source data 1* and *2*). Below: maximum life span (logarithm of days) in six model organisms, data from the AnAge longevity database (http://genomics.senescence.info/species/). (**B**) Heatmap of ranked relative abundance of OTU composition at the phylum-level for different model organisms (*Figure 1—source data 1*). Bacterial phyla are ordered by their relative abundance in the TK. The ranked abundance is color-coded, with higher ranks (red) indicating greater relative abundance and lower ranks (blue) indicating lower relative abundance. White cells mark phyla that are not detected in the respective model organism.

The following source data is available for figure 1:

**Source data 1.** Ranked abundance of bacterial phyla shared among turquoise killifish (TK), zebrafish, mouse and human (Materials and methods).
**Source data 2.** Relative phylum abundance of bacterial phyla shared among turquoise killifish (TK), zebrafish, mouse and human (Materials and methods).

diversity indexes and different recorded ecological factors, including altitude, pond size, maximum pond depth, vegetation, water conductivity, water temperature and water turbidity (*Figure 2—source data 2*). None of the recorded factors was significantly correlated to alpha diversity levels (data not shown). This suggests that other parameters, such as food availability, fish genetics, and presence of parasites or additional biotic or abiotic factors might be the key determinants of fish GM composition in the wild.

Although microbiota diversity in wild populations was higher than in laboratory-raised fish, the microbial composition of laboratory fish was not separated from the species-specific bacterial composition in wild fish, and was contained within the natural microbial variation of the species (*Figure 2D*). Importantly, one wild population (M1) had a divergent composition from all other populations, possibly due to over-representation of Planctomycetes (*Figure 2B and D*, *Figure 2—figure supplement 1C and E*). Within-group diversity measures (beta diversity) showed that, while laboratory fish's bacterial diversity was lower than in the wild populations, between-group bacterial diversity was higher between population M1 and all the other populations, including laboratory fish (*Figure 2E–F*). These results indicate that population M1 had a more divergent microbial diversity of all the tested populations, and that the laboratory fish were not an outlier group (*Figure 2D*). Based on this, laboratory fish share a core microbiota with wild populations, despite the differences in culturing conditions between the laboratory and the wild, similar to what is seen in zebrafish (*Roeselers et al., 2011*). Thus, while wild turquoise killifish populations differ from one another in terms of microbial composition – possibly in association with ecological differences among different

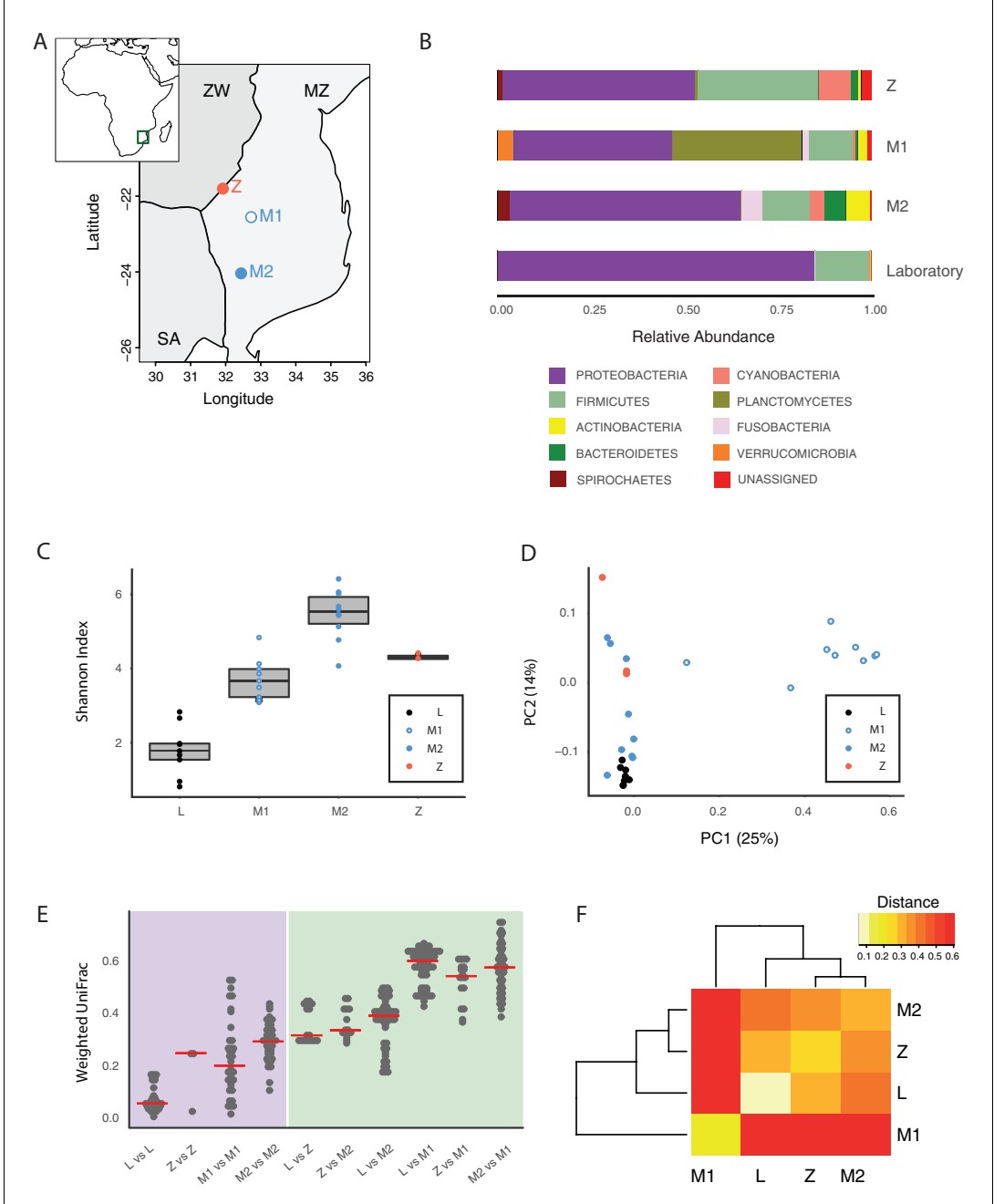

**Figure 2.** Core microbiota is conserved between wild and captive TK. (**A**) Map location of wild populations collected in Zimbabwe (Z) and Mozambique (M1 and M2). ZW: Zimbabwe; MZ: Mozambique; SA: South Africa. (**B**) Relative abundance of bacterial phyla in wild populations and laboratory fish. (**C**) Shannon Index alpha diversity in laboratory and wild fish (L = laboratory, derived from individuals originally collected in Zimbabwe). (**D**) PCoA of the Weighted UniFrac beta diversity distance for wild and laboratory fish. Adonis test: L vs. all groups: p value < 0.001, M2 vs. M1: p value < 0.001, Z vs. M1 and M2: p value < 0.05. (**E**) Dotplot of the Weighted UniFrac distance values to visualize beta diversity within (purple background) and between (green background) the different populations. Single dots represent comparisons between individual fish. Red horizontal line indicates the median for each comparison. (**F**) Heatmap of the Weighted UniFrac distance among wild populations and laboratory fish.

The following source data and figure supplement are available for figure 2:

**Source data 1.** Collection points of the wild fish populations.

**Source data 2.** Ecological factors associated with wild fish populations.

*Figure 2 continued on next page*

*Figure 2 continued*

**Figure supplement 1.** Bacterial diversity measures in captive and wild turquoise killifish populations.

localities – laboratory fish recapitulate the core gut microbiota of wild populations, making them ideally suited to study how a complex microbiota influences host physiology.

## Aging in the TK is associated with loss of microbial diversity and decreased expression of genetic markers of gut health

An important question is what changes in the gut microbial population occur during aging in vertebrates. To gain insight into the changes in GM composition occurring throughout aging in the TK, we performed 16S rRNA gene amplicon surveys in the gastrointestinal tract of 6-week-old young-adult and 16-week-old individuals raised in captivity (*Figure 3A*, n = 16 young and 14 old fish). We found that bacterial taxonomic diversity (alpha diversity) significantly decreased between 6-week-old and 16-week-old fish, while bacterial abundance measured from stool was not changed (*Figure 3B*, *Figure 3—figure supplement 1A and B*). This indicates that, similar to humans (*Claesson et al., 2012*), aging in the TK is characterized by a significant reduction in gut bacterial richness. Bacterial composition was also significantly altered between young and old fish guts (*Figure 3C*), including an age-dependent decrease in Firmicutes (Kruskal-Wallis test, p value = 0.002) and Actinobacteria (Kruskal-Wallis test, p value = 0.013). Young and old fish GM had significantly different community structures (*Figure 3C*, Unweighted UniFrac p value < 0.001; Bray-Curtis p value < 0.001, Adonis test, *Figure 3—figure supplement 1C*). While individual fish GM diversity was higher in young than in old fish (*Figure 3B*), differences among old fish GM were more pronounced compared to young fish (Unweighted UniFrac beta diversity, Bonferroni-corrected p value = 6.7E-06; Bray-Curtis p value = 0.003; *Figure 3—figure supplement 1D and E*). This indicates that although aging in the TK was associated with decreased bacterial taxonomic diversity within each fish's gut, distinct old fish guts had highly divergent bacterial communities. While young fish guts were significantly enriched for Bacteroidetes, Firmicutes and Actinobacteria, old fish were dominated by Proteobacteria (*Figure 3D*). We next investigated the predicted functional metagenome biomarkers associated with young and old fish's guts using PICRUSt (*Langille et al., 2013*) and LEfSe (*Segata et al., 2011*). While young fish had GM associated with glycolysis and polysaccharide metabolism, old fish's GM was depleted of bacteria associated with carbohydrate, nucleotide and amino acid metabolism, and was enriched for bacteria associated with pathogenesis, transport and catabolism (*Figure 3E*). In particular, bacterial motility and flagellar assembly was strongly increased in GM from old fish. These terms are associated with increased virulence in bacteria (*Josenhans and Suerbaum, 2002*), supporting that old fish had a higher prevalence of potentially pathogenic bacteria.

To examine whether metagenomic biomarkers from bacterial taxonomic diversity (i.e. OTU) data were consistent with host responses, we asked which were the transcriptional changes in the gut associated with young and old status. To this end, we performed an RNA-Seq analysis of whole gut in four 6-week-old and four 16-week-old fish (*Figure 3F and G*, *Figure 3—figure supplement 2* and Materials and methods). Young fish had a distinct expression signature of active proliferation (*Figure 3G*, *Figure 3—figure supplement 2*), consistent with the bacterial metagenomic signature of replication. Old fish, on the other hand, had significant Gene Ontology terms associated with immune and defense responses against pathogens as well as inflammation, consistent with the bacterial metagenome signatures associated with host disease and overall virulence. Together, these results strongly support that in old individuals both changes in bacterial composition and gut transcriptome are consistent with a markedly pathological gut environment, while young fish are characterized by a molecular signature of healthy gut and a commensal bacterial community.

## Young GM transfer prolongs life span and delays age-dependent motor decline

Interventions aimed at directly modifying the complex microbial composition in experimental organisms and patients have been mostly focused on treating diseases such as *Clostridium difficile* infections (*Dodin and Katz, 2014*; *Lee et al., 2016*), as well as obesity and type two diabetes

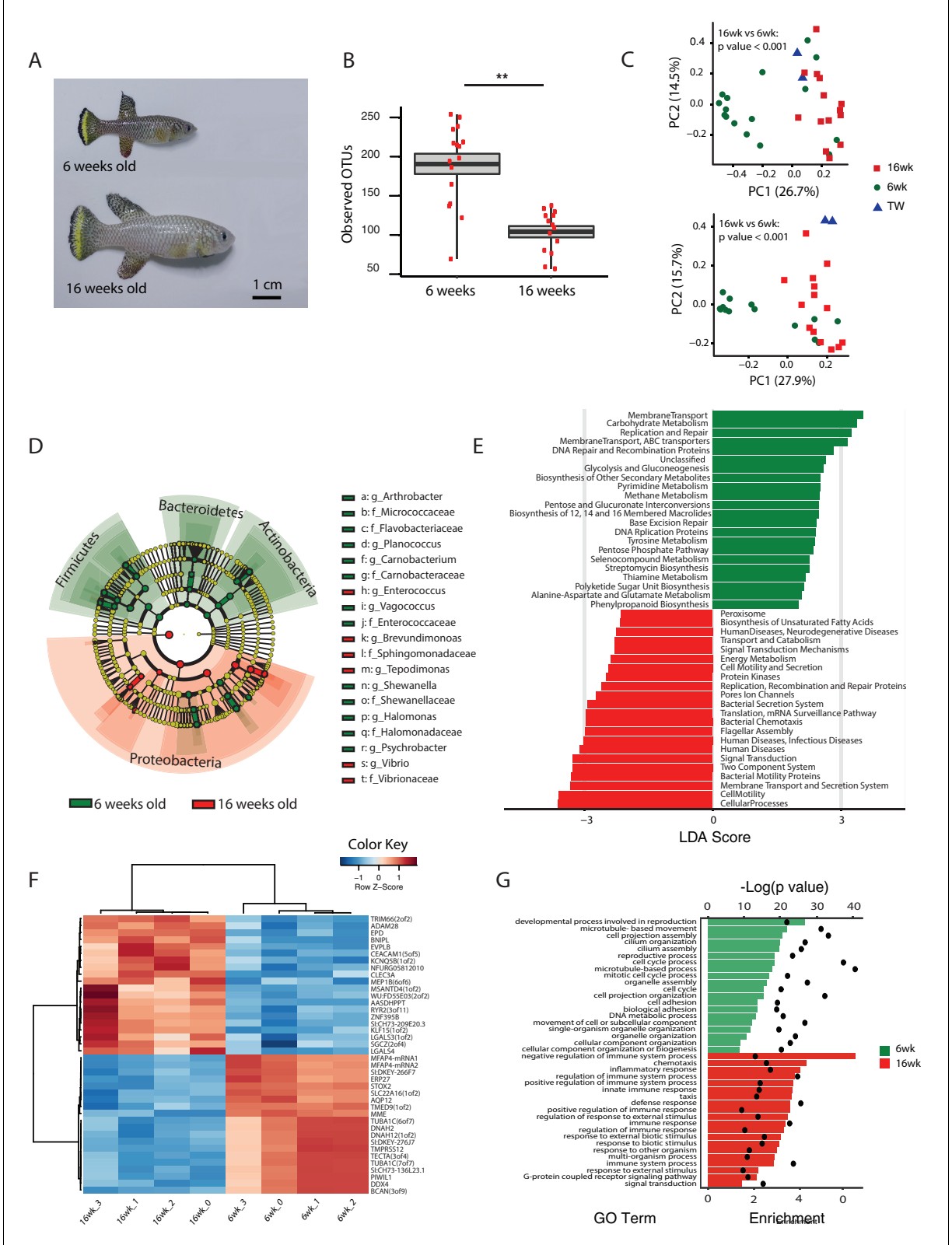

**Figure 3.** Changes in GM and gut transcriptome between young and old fish. (**A**) Representative 6-week-old (young) and 16-week-old (old) male TK. (**B**) Alpha diversity changes in observed OTUs in young (6 weeks) and old (16 weeks) TK. N = 16 6-week-old fish and 14 16-week-old fish. The groups are compared using the Mann Whitney U test; ** indicates a p value < 0.001. (**C**) Beta diversity microbiota analysis separates samples based on age. Above: Bray-Curtis analysis, below: Unweighted UniFrac analysis. TW: tank water control. Adonis test, p value < 0.001 in both comparisons between 6-

*Figure 3 continued on next page*

*Figure 3 continued*

week and 16-week-old fish. (D) Cladogram representing microbial taxa enriched in young (green) versus old (red) individuals. (E) Predicted metagenome function in young (green) and old (red) groups (LEfSe), representing functions with p value < 0.001. The x-axis indicates the Linear Discriminant Analysis score for all the significant metabolic functions. (F) Expression heatmap for the twenty top differentially expressed genes (DEGs) between young and old fish (n = 4 for each group). Blue to red color represent low to high expression. Top 20 genes are highly expressed in old fish, the bottom 20 are highly expressed in young fish. (G) Top 20 Gene Ontology (GO) terms of the DEGs between young and old fish. Enrichment values (bars) and the negative natural logarithm of p values (black dots) are shown.

The following figure supplements are available for figure 3:

**Figure supplement 1.** Aging in the GM: bacterial diversity.

**Figure supplement 2.** RNA-Seq analysis of young and old fish intestines.

(*Kootte et al., 2012*; *Turnbaugh et al., 2006*). However, the impact of a young GM in modulating aging and life span has not been explored to date in vertebrates (*Clark et al., 2015*). To test whether resetting a young-like GM in middle-age could impact aging and affect life span, we treated middle-age fish (9.5-week-old) overnight with an antibiotic cocktail (VMNA, i.e. vancomycin, metronidazole, neomycin, ampicillin) (*Figure 4A*, *Figure 4—figure supplement 1A* and Materials and methods). The antibiotic treatment significantly reduced gut microbial content compared to pretreatment levels (*Figure 4—figure supplement 1B*). Antibiotic-treated fish were then exposed for 12 hr to the following conditions: 6-week-old donor fish gut content (Ymt), 9.5-week-old fish gut content (Omt) and sham (Abx) (*Figure 4A* and *Figure 4—figure supplement 1A*). After antibiotic treatment and 12 hr acute gut recolonization, fish were reintroduced in the water recirculation system in individual tanks and were subjected to regular feeding (Materials and methods). Their survival under the different experimental conditions was then scored (*Figure 4—source data 1* and *Figure 4—figure supplement 1D* for the replicates of the survival experiments). Ymt fish underwent dramatic life span prolongation compared to three control groups, which received: (i) antibiotic-only (Abx) (21% life span increase in median life span, Logrank test p value = 5.89E-05), (ii) antibiotics and same-age (i.e. 9.5 weeks) gut content (Omt) (41% increase in median life span, Logrank test p value = 5.08E-06), or (iii) no-treatment (wt) (37% increase in median life span, Logrank test p value = 4.04E-09) (*Figure 4B* and *Figure 4—figure supplement 1D*). Noteworthy, acute antibiotic treatment alone was sufficient to increase fish life span compared to the wt group (14% median life span increase, Logrank test p value = 0.0129) (*Figure 4B*). However, Omt fish did not live longer than the control, wt group (*Figure 4B*). Since Abx outlived Omt and wild-type fish, while Ymt fish outlived Abx fish, it is plausible that middle-age GM composition might be primed to induce damage in the host and that its removal is therefore beneficial. However, as the recolonization of middle-age individuals with young fish gut content after antibiotic treatment prolongs life span even more, this implies that young GM, per se, has beneficial effects on host physiology that are additive to the effects of the antibiotic treatment.

We then asked whether treating young fish with old fish GM could also affect life span. Compared to fish that received either the same-age GM or to untreated control fish, 6-week-old fish receiving 16-week-old fish's GM after antibiotic treatment did not have a different life span (*Figure 4—figure supplement 1C*). Additionally, unlike middle-age fish treated with antibiotics, young fish receiving antibiotic treatment did not live longer than untreated control fish (*Figure 4—figure supplement 1C*, *Figure 4—source data 2*). These results suggest that the timing of GM transfer is critical to inducing systemic effects and modulating life span.

It was shown in previous work that spontaneous exploratory behavior in TK decreases with age (*Genade et al., 2005*; *Valenzano et al., 2006b*). We therefore asked whether treating middle-age fish with young fish GM after antibiotic treatment could improve exploratory behavior performance, considered as an integrated measure of individual health. Using an automatic video-tracking system (Materials and methods), we assayed spontaneous locomotor activity. Young, 6-week-old fish, were significantly more active than 16-week-old fish (*Figure 4—source data 3* and *Figure 4C*, Kruskal-Wallis chi-squared = 10.752, df = 1, p value = 0.00104). Remarkably, Ymt were more active at 16 weeks of life than Omt and wt fish at the same age, resembling younger fish performance

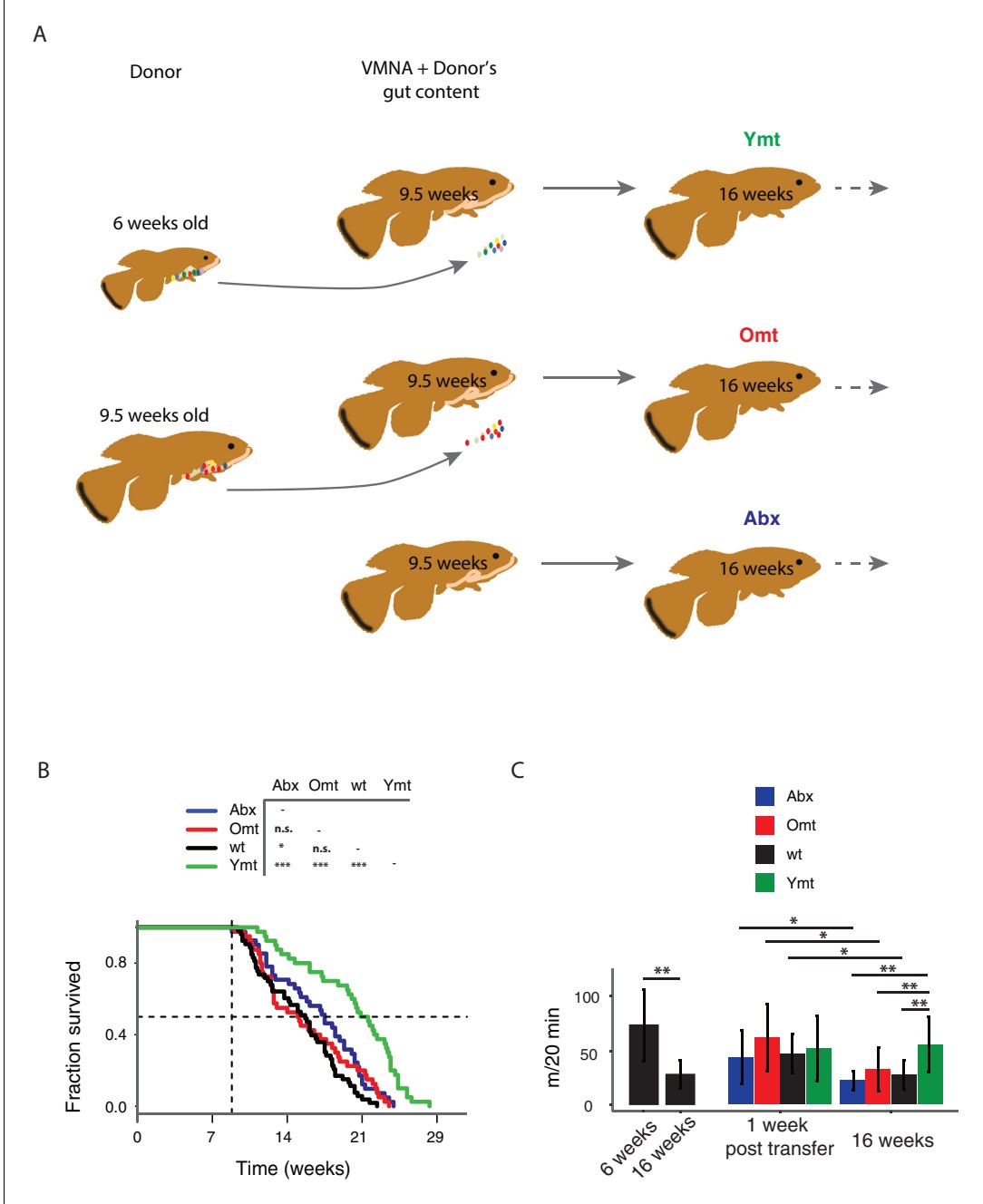

**Figure 4.** Transferring young GM to adult fish prolongs life span and delays motor decline. (**A**) Schematic representation of the microbial transfer experiment (Materials and methods). Experimental group legend, Abx: fish receiving only antibiotic treatment at 9.5 weeks without direct recolonization. Omt: fish receiving same-age GM transfer after antibiotic treatment at 9.5 weeks. Wt: wild-type, untreated fish. Ymt: fish receiving 6-week-old fish GM transfer after antibiotic treatment at 9.5 weeks. VMNA: antibiotic cocktail of vancomycin, metronidazole, neomycin and ampicillin. (**B**) Survival analysis. Statistical significance is calculated by Logrank test. * indicates a p value < 0.05; *** indicates a p value < 0.001. (**C**) Exploratory behavior in different treatments. Y-axis indicates average distance (in meters) covered in 20 min. Young and old wild-types are compared with a Kruskal-Wallis test (left), the remaining groups are compared using a Dunn Kruskal-Wallis test for multiple comparisons, and the p values are adjusted based on BH correction. Statistical significance: * indicates a p value < 0.05; ** indicates a p value < 0.01; *** indicates a p value < 0.001.

The following source data and figure supplement are available for figure 4:

**Source data 1.** Survival data for transfer experiment after antibiotic treatment at 9.5 weeks of age.

**Source data 2.** Survival data for transfer experiment after antibiotic treatment at 6 weeks of age.

*Figure 4 continued on next page*

*Figure 4 continued*

**Source data 3.** Spontaneous locomotor activity (cm/20 min).

**Figure supplement 1.** Experimental design and consequences of gut microbiota transfer protocol.

(*Figure 4C*, Dunn Kruskal-Wallis multiple comparison test, BH-adjusted p value = 0.004). Additionally, all groups except Ymt underwent a significant decrease in spontaneous locomotor activity from a week post transfer to 16 weeks of life. This suggests that the transfer of young fish gut content had long-lasting effects on a global measure of physiological health, influencing individual survival and spontaneous exploratory behavior. Thus, depleting middle-age individuals from their resident GM was beneficial when acutely recolonized by young-associated GM (Ymt), and in part also when it was not followed by any acute recolonization (Abx). On the other hand, acutely recolonizing the gut with same age GM after antibiotic treatment did not lead to differences compared to the untreated control group. These results establish gut microbial recolonization as a powerful life span enhancing intervention, which leads to significant effects also on behavioral performance.

## Acute GM transfer affects microbial composition at old age

To assess the extent to which one acute transfer reset the GM in recipient fish after VMNA antibiotic treatment, we performed 16S rRNA gene amplicon surveys in fish that underwent microbial transfer at 9.5 weeks of age. Untreated 6-week-old (6wk) and 16-week-old (16wk) fish were also included in the analysis. One week post transfer there were no significant differences in alpha diversity metrics among treatment groups (*Figure 5—figure supplement 1A and B*). Remarkably, the GM of 10-week-old wild-type fish (10wk) was intermediate between 6wk and 16wk fish (*Figure 5—figure supplement 1A–D*). However, at one-week post transfer, Ymt and Omt fish had already significantly different gut microbial population diversity (Unweighted UniFrac and Bray-Curtis distances, *Figure 5—figure supplement 1C and D* and *Figure 5—source data 1*). Seven weeks after the microbial transfer (16 weeks of age, corresponding to the median life span for this species in captivity), Ymt fish had significantly higher bacterial richness compared to wild-type, 16-week-old fish (*Figure 5A*, Dunn Kruskal-Wallis test, BH-adjusted p value = 0.009). This shows that one acute transfer had long-lasting effects on GM diversity. Bacterial OTU abundance at 16 weeks was higher in Ymt fish compared to Omt fish (*Figure 5A*), but lower than 6-week-old wild-type fish (6wk).

GM community structure in Ymt fish was also significantly altered compared to Omt and 16wk (*Figure 5B* and *Figure 5—figure supplement 1E*); however, it did not statistically differ from young wild-type fish, i.e. the 6wk group. Furthermore, based on hierarchical clustering, Ymt fish clustered preferentially with 6wk fish (*Figure 5C*), showing that the GM-transfer from young donors significantly reset a young-like GM.

Young fish (6wk), as well as fish treated with young GM (Ymt), were more enriched with members of the Bacteroidetes and Firmicutes (*Figure 5D*) and with the genera *Carnobacterium*, *Arthrobacter*, *Exiguobacterium*, *Planococcus*, *Psychrobacter*, *Enterococcus* and *Halomonas* (*Figure 5E*). Additionally, analyzing the correlation between bacterial genera abundance and the group-specific median lifespan, we identified a set of bacterial genera whose abundance is highly correlated with group-specific longevity (*Figure 5—source data 2*).

We then used predicted functional metagenome analysis from GM composition to compare experimental fish groups receiving same-age GM transfer and no-treatment controls to young fish and fish receiving young gut content. The latter group was enriched for increased saccharolytic potential, DNA repair and recombination among other functions (*Figure 5—figure supplement 1F*), which are functional terms associated with a younger, healthy-like physiological state. Together, these results show that gut microbial recolonization from young killifish resulted in increased microbial diversity and a persistence of a young-like bacterial community until old age, whose composition could be at least in part responsible for the life span prolongation.

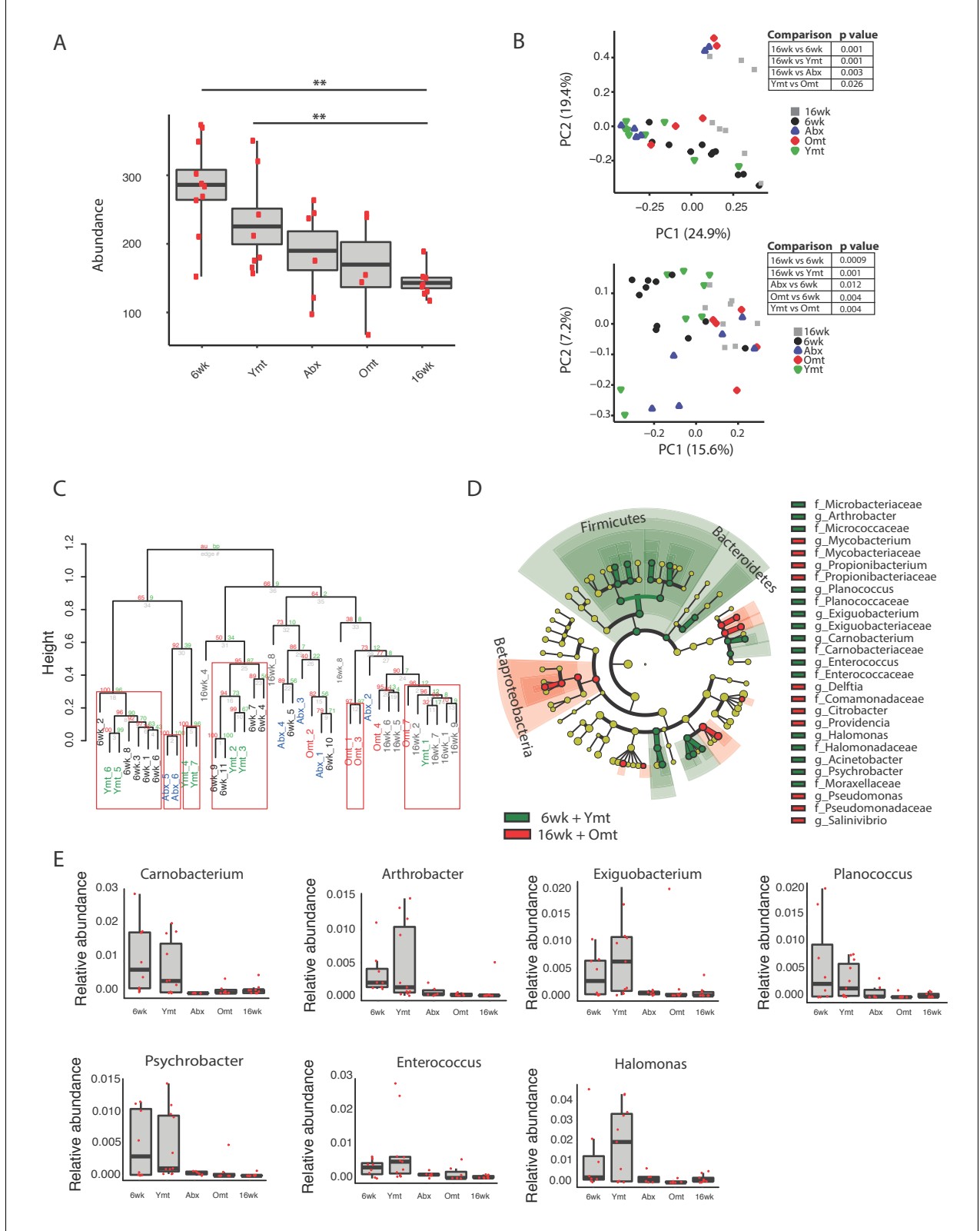

**Figure 5.** Microbiota transfer at 9.5 weeks influences microbial composition at 16 weeks. (**A**) Alpha diversity measured by number of observed OTUs in experimental groups at 16 weeks (16wk, Abx, Omt, Ymt) and young controls (6wk). Dunn Kruskal-Wallis test for multiple comparisons. The p values are adjusted based on BH correction. Statistical significance: ** indicates a p value < 0.01. (**B**) Microbiota community analysis using Bray-Curtis (above) and Unweighted UniFrac (below) separates samples based on young vs. old GM treatment (significant p values are shown, Adonis test). (**C**) Hierarchical

*Figure 5 continued*

clustering on OTU data. Significant clusters are highlighted by red rectangles. P values: au = approximately unbiased; bp: bootstrap probability. (**D**) Cladogram of bacterial taxa enriched in young wild-type (6wk) and Ymt groups versus 16wk and Omt. (**E**) Boxplots of selected bacterial genera relative abundance in experimental groups. Bacterial genera are those represented in *Figure 5D*, filtered by r-square value larger than 0.8, based on the correlation between bacterial genus abundance vs. median life span in each individual group (*Figure 5—source data 2*).

The following source data and figure supplement are available for figure 5:

**Source data 1.** Beta diversity significance at one-week post-transfer.
**Source data 2.** Regression between lifespan and genus abundance.
**Figure supplement 1.** Effects of gut microbiota transfer on OTU composition and metagenome.

## Microbial transfer determines bacterium-to-bacterium connectivity at old age

To investigate whether gut recolonization affected bacterium-to-bacterium association in different experimental groups, we harnessed OTU co-occurrence to generate bacterial connectivity networks (*Agler et al., 2016*; *Biagi et al., 2016*). Significantly co-occurring genera within each group composed a network, whose nodes were single bacterial genera. Bacterium-to-bacterium connections (edges) were established based on r-square values (Materials and methods). Using this analysis, we found that Proteobacteria had higher connectivity in the shorter-lived 16wk and Omt groups compared to 6wk, Ymt and Abx (*Figure 6A*). Overall, 6wk and Ymt fish had the largest networks and a higher number of highly-connected nodes (hub nodes), showing that a few genera of bacteria significantly co-occurred with a large set of other bacterial genera (*Figure 6B and C*).

We then tested whether the relation between age of the host and bacterial connectivity observed in TK guts was also conserved in mammals. To this end, we analyzed a published mouse 16S amplicon survey from an aging cohort (*Langille et al., 2014*). Similar to what we found in TK, young mice GM had more significantly connected nodes and a higher number of bacterial hubs compared to middle-age and old mice (*Figure 6B*, inset).

The identification of wild-type specific young and old-related hub bacterial genera enabled us to study whether Ymt, Omt and Abx shared hub bacterial genera with either young or old fish groups. Two hub bacterial clusters were identified in Ymt fish (Materials and methods). Strikingly, one was composed of bacterial genera that were also hub-bacteria in young wild-type fish (6wk) and included *Exiguobacterium*, *Planococcus*, *Propionigenium* and *Psychrobacter* (*Figure 6C*, Ymt network, green nodes), while the other was composed of hub bacteria from old wild-type fish (16wk) hub and included *Propionibacterium*, *Delftia*, and *Citrobacter* (*Figure 6C* and *Figure 6—source data 1*). Remarkably, the bacterial hubs identified in Omt and Abx overlapped exclusively with the old wild-type group (16wk) (*Figure 6C*, orange nodes, and *Figure 6—source data 1*). These results support that bacterial network topology reflects host age both in fish and mice, with younger biological age associated with larger networks. Acute microbiota transfers significantly affect gut microbial population topology and subsets of bacterial genera are identified as hub nodes in young-like and old-like microbial communities. Therefore, not only bacterial composition, but also bacterium-bacterium co-occurrence carries a key signature of host life span and network topology depends on a few key bacterial hubs.

## Microbial transfer affects expression of host genes associated with defense to bacteria, Tor pathway and extracellular matrix

Because acute microbial transfer dramatically changed GM composition by resetting a young-like microbial community in Ymt fish, we asked whether young GM transfer could also reset the host transcriptome towards a young-like status. To this end, we performed host intestine RNA-Seq in wild-type young (6wk) and old (16wk) fish, as well as in Omt and Ymt fish at 16 weeks of age (*Figure 7A and B*, *Figure 7—figure supplement 1* and Materials and methods). Interestingly, while young GM transfer reset a young-like GM environment in aged fish, host gene transcripts separated based on fish age, and all the transcriptomes obtained from 16 week old fish groups (16wk, Omt,

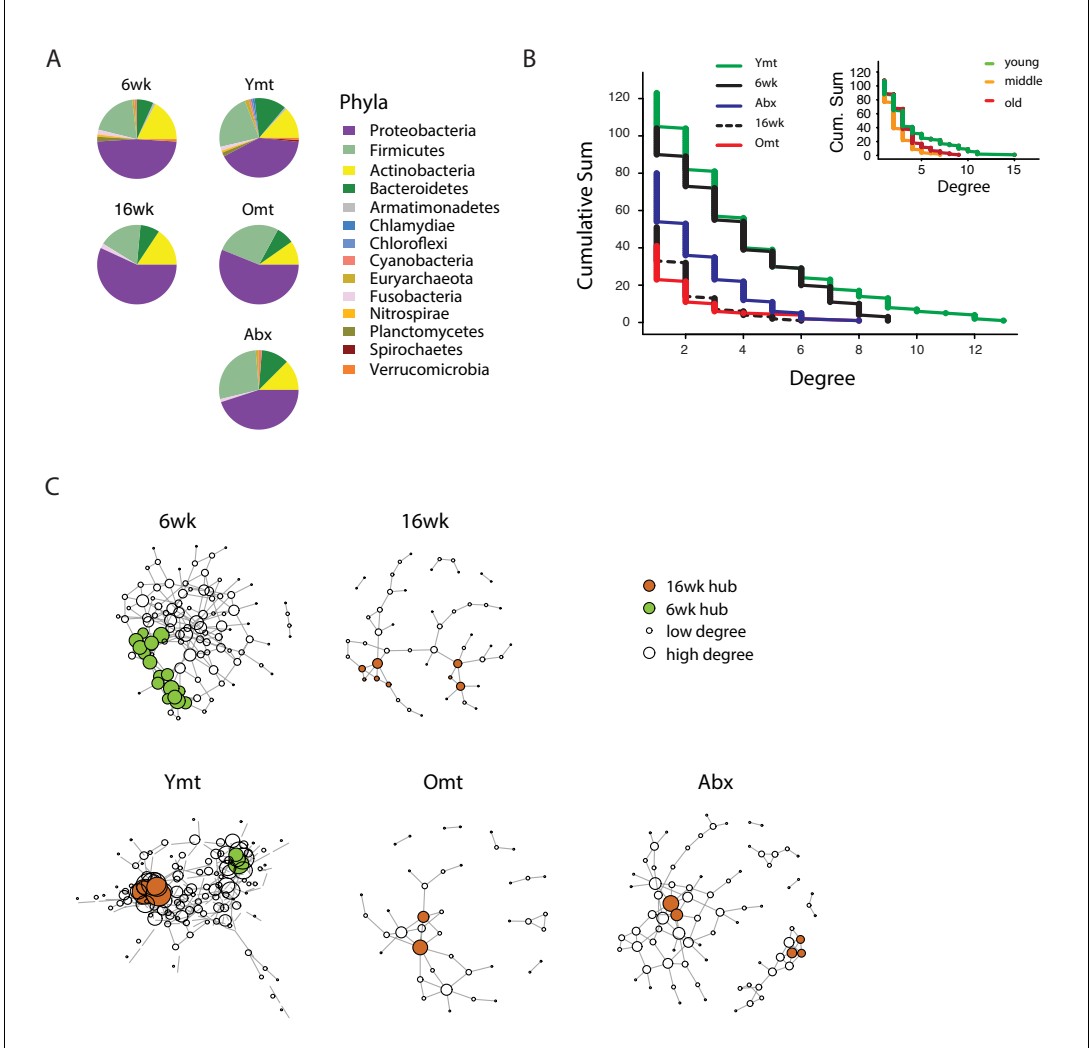

**Figure 6.** Bacterial co-occurrence connectivity. (**A**) Phyla composition of the bacterial co-occurrence networks in experimental groups at 16 weeks (16wk, Ymt, Omt, Abx) and at 6 weeks (6wk) (Materials and methods). (**B**) Cumulative sum of degree distribution in co-occurrence networks of all analyzed TK experimental groups (black: 6wk; dashed black line: 16wk; green: Ymt; red: Omt; blue: Abx) and a mouse cohort from (*Langille et al., 2014*) (inset, green: young mice; orange: middle-age mice; red: old mice). X-axis shows the degree count and y-axis the cumulative sum of nodes with the corresponding degree. (**C**) Visualization of co-occurrence networks in Fruchtermann-Reingold layout of 6wk, 16wk, Ymt, Omt, and Abx. Circle size increases with degree count and circle color corresponds to 6wk hubs (green) or 16wk hubs (orange).

The following source data is available for figure 6:

**Source data 1.** Network hubs of OTU-based networks.

Ymt) clustered together (*Figure 7A*). However, comparing transcriptomes in Omt and Ymt revealed a distinct signature of defense response to bacteria in Ymt and increased expression of genes associated with hyaluronic acid metabolism in Omt (*Figure 7B and C*). Comparing the expression levels of all groups to the 6wk group, we identified a set of genes whose expression in the gut was significantly changed comparing 16wk and Omt with 6wk; comparing 16wk and Ymt, but unchanged between Ymt and 6wk (Fisher exact test, Benjamini-Hochberg FDR = 0.1). These were genes associated with the TOR-pathway (DEPTOR) and with cell adhesion and extracellular matrix composition (DSCAM) (*Figure 7D*), suggesting a potential difference in cell adhesion and gut permeability between 16wk and Omt from one side, and Ymt and 6wk on the other side. Since we generated OTU abundance tables for individual fish and we also sequenced intestinal transcripts for the same

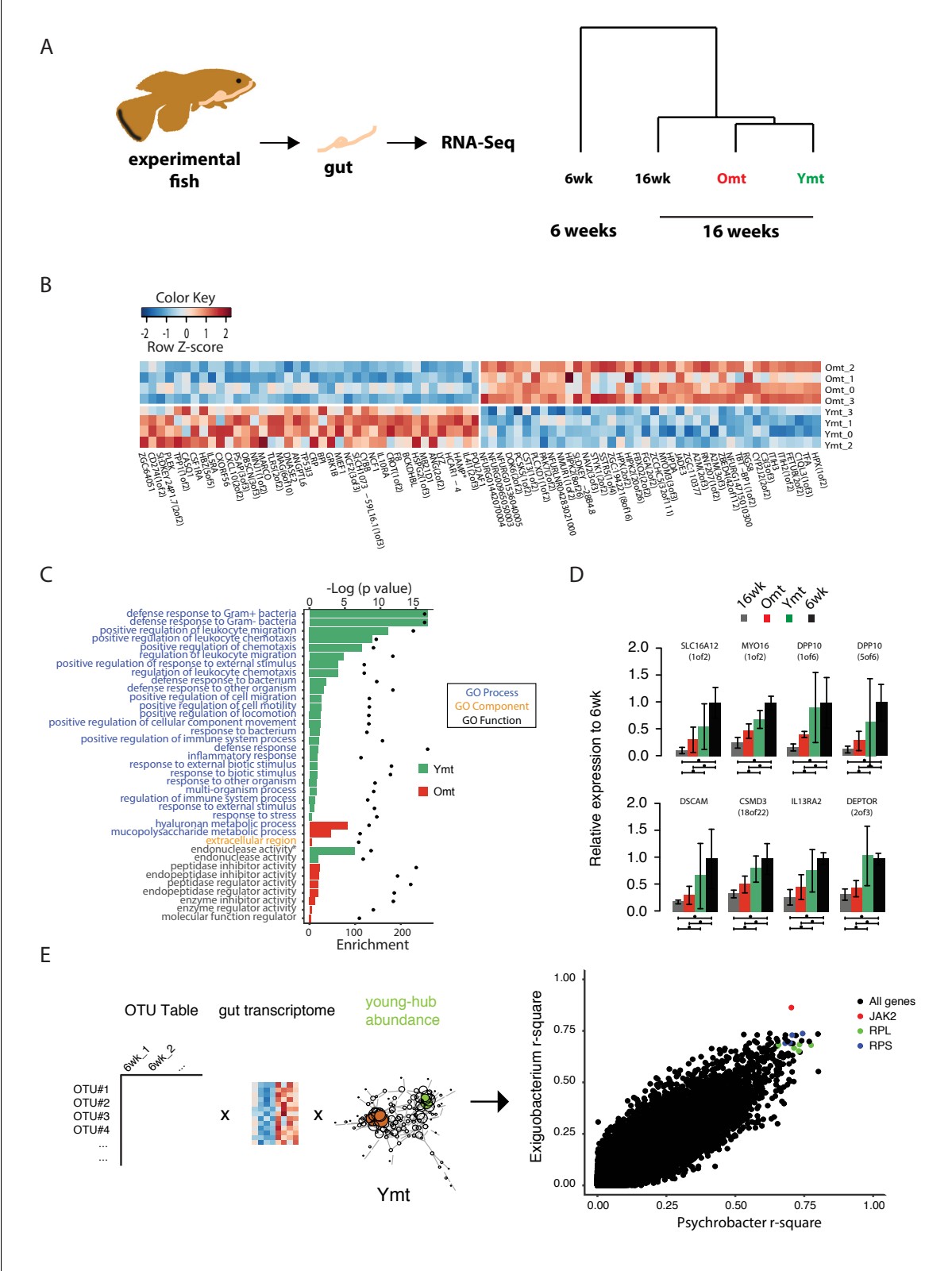

**Figure 7.** Transcriptional changes in host intestine after acute GM transfer. (**A**) Cluster analysis of gut RNA-Seq from intestines from 6-week-old fish (6wk) and 16-week-old fish (16wk, Omt (red) and Ymt (green)). (**B**) Expression heatmap for the 80 top differentially expressed genes (DEGs) between Ymt and Omt fish (n = 4 for each group). *HAMP is the best protein blast hit in *Danio rerio* of the TK gene NFURG05812010005 (**Figure 7—source data 1**). (**C**) Gene Ontology (GO) analysis of the DEGs between Ymt and Omt fish. Enrichment values (bars) and the negative natural logarithm of p

*Figure 7 continued on next page*

*Figure 7 continued*

values (black dots) are shown. *Endonuclease activity** indicates the following GO function: 'Endonuclease activity, active either with ribo or deoxyribonucleic acids and producing 3' phosphomonoesters'. (**D**) Relative gene expression levels compared to 6-week-old fish (6wk) of genes significantly differentially expressed between 16wk (gray), Omt (red) and 6wk (black), but not between Ymt (green) and 6wk. These genes are also significantly differentially expressed between 16wk and Ymt. (**E**) Based on the correlation matrix between OTU abundance and transcript levels in individual fish (*Figure 7—source data 2*), we identified the gene transcripts whose expression correlates more significantly with the hub bacteria shared between Ymt and 6wk fish (full analysis in *Figure 7—figure supplement 2*). Displayed are the top hits (chosen from an r-square value larger than 0.65 on both axes) shared between *Psychrobacter* and *Exiguobacterium*, which have the highest OTU-transcript correlation trend based on *Figure 7—figure supplement 2*. RPLs include RPL7A, RPL8, RPL12, RPL37, RPL14, MRPL35; while RPS include RPS3A, RPS7, RPS11, RPS24.

The following source data and figure supplements are available for figure 7:

**Source data 1.** Best hits from zebrafish pBlast on DEGs not annotated in the *Valenzano et al., 2015* genome paper.

**Source data 2.** Bacterial OTU to host transcripts correlation matrix.

**Figure supplement 1.** Volcano plots from the RNA-Seq data in the experimental groups.

**Figure supplement 2.** OTU-transcript correlations between hub-genera.

individuals, we then generated an OTU-to-host transcripts correlation matrix (*Figure 7—source data 2*). Taking advantage of this resource, we could identify the transcripts that were highly correlated with the hub genera that are shared between Ymt and young fish (6wk) networks (*Figure 6C*, *Figure 7E* and *Figure 7—figure supplement 2*). Remarkably, these genes include Jak2, an important gene involved in cellular proliferation and differentiation, and several genes encoding S and L-ribosomal proteins (RPS and RPL genes) (*Figure 7—figure supplement 2*), whose expression in other species has been strongly associated with aging and longevity (*Steffen et al., 2008*). Thus, while transfer of young GM to middle-age fish maintained a young-like GM community throughout old age, the gene-expression signature associated with host gut at 16 weeks of age did not indicate overall rejuvenation. However, enhanced defense response against bacteria in Ymt compared to Omt is compatible with an increased capacity to resist to the attack of pathogenic gut bacteria, which could provide the basis for longer life span. Remarkably, hyaluronic acid metabolism, altered between the experimental groups receiving young and old gut contents, has been associated with increased inflammation, deregulated immune response and risk for cancer (*Cho et al., 2017*; *Tian et al., 2013*), all of which could provide the basis for life span modulation. Finally, we provide a dataset that allows to associating OTU levels in individual fish to intestinal transcripts, enabling to investigate on how GM and host gene expression are mutually regulated. This resource suggests that transfer of young gut microbiota to middle-age individuals might be affecting host gut proliferation and differentiation, as well as ribosomal biogenesis.

## Discussion

Key aims of research on aging are to understand the molecular mechanisms behind the phenotypic changes that occur during aging and to identify novel life span enhancing interventions. Using the TK as a naturally short-lived vertebrate model system, we report the characterization of the changes in GM composition occurring during aging and the discovery of a novel life span enhancing intervention achieved by acutely transferring young GM to middle-age individuals after antibiotic treatment. This intervention resulted in the maintenance of an overall healthier physiological status, a highly diverse and young-like gut microbial community at late age and in an enhanced transcriptional signature of defense responses to bacteria.

Our results show that TK are characterized by a complex GM community, more species-rich than worms and flies in laboratory conditions and of the same order of magnitude of mammals, both in abundance and composition. Indeed, the four most abundant bacterial phyla observed in the TK are also the four most abundant phyla found in humans and mice. However, unlike mammals, Proteobacteria is the most abundant gut bacterial phylum found in killifish, similarly to other aquatic species, such as zebrafish (*Roeselers et al., 2011*). Since it is the shortest-lived vertebrate to date

reproduced in captivity, the TK can become an ideal model to dissect the links between GM diversity and host aging.

Intriguingly, we show that while the gut bacterial diversity of captive-raised TK is lower than that of wild-caught populations, captive fish still recapitulate the core microbial diversity of wild TK. Laboratory-raised fish are not represented by distinct gut bacterial communities, but are rather dominated by a few high-abundance bacterial taxa that are already present in wild killifish populations. This supports the conclusion that fish raised in captivity have a GM community that is representative for the species in nature. Additionally, we found that wild populations of TK show large between-population gut bacterial diversity, with the emergence of population-specific high-abundance taxa. This is plausibly caused by locality-specific differences in ecological conditions, including climate, soil composition, parasites and diversity of food sources (*Nezhybová et al., 2017*; *Reichard et al., 2017*).

Analyzing the changes in GM composition between young and old fish, we found that young fish are characterized by a large taxonomic bacterial diversity. Old fish are less OTU-rich, similar to what is observed in human cohorts from different age classes (*Claesson et al., 2012*). Intriguingly, while individual bacterial diversity (alpha diversity) is higher in young individuals, old fish are more dissimilar from one another, that is, while each individual old fish has a more homogeneous GM composition than young fish, any two old individuals have more divergent bacterial communities compared to young individuals. This result raises the possibility that the changes in composition and relative abundance in GM communities from young to old individuals could be a function of (i) initial individual GM composition, (ii) differences in individual immune system composition and function or (iii) a combination of initial individual GM composition and host immune function.

Aging in this experimental model was not only associated with reduced bacterial species richness, but more specifically with loss of bacterial taxonomic units involved in carbohydrate, nucleotide and amino acid metabolism, which in mice are associated with aging (*Langille et al., 2014*). These same changes have been associated in humans with unhealthy aging (*Claesson et al., 2012*; *Rampelli et al., 2013*), as well as with chronic conditions such as obesity, type two diabetes and insulin resistance (*Neis et al., 2015*). The shift in microbial composition between young and old fish was consistently characterized by a higher prevalence of Proteobacteria in old individuals, while young individuals were significantly more enriched in Firmicutes, Actinobacteria and Bacteroidetes. Additionally, functional metagenome analysis showed that young fish had GM associated with carbohydrate metabolism, replication and repair, as well as DNA repair, indicating the young GM's ability to protect itself against assault and thus maintain homeostasis. On the other hand, bacterial pathogenicity in the gut is associated with the accumulation of mutations over time, induced by the failed capacity to repair them (*Leimbach et al., 2013*). Old GM was enriched in potentially pathogenic bacteria, associated with dysbiosis. Additionally, functional metagenome analysis on the bacterial communities present in old fish guts indeed found them to be associated with host disease. Consistently, while young gut transcriptomes were associated with high expression of host genes involved in cell cycle activity, likely associated with proliferation and differentiation, old gut transcriptomes were associated with host immune responses to pathogenic bacteria, reflecting the prevalence of more pathogenic bacterial taxa.

Although GM transfers from young, healthy donors have found applications in the clinic to treat acute gut infections such as those associated with *Clostridium difficile* (*Lee et al., 2016*) and have been proposed to treat obesity, metabolic syndrome and even neurodegenerative diseases (*Marotz and Zarrinpar, 2016*; *Xu et al., 2015*), the application of this methodology as an anti-aging intervention has not been explored to date. Remarkably, despite single associations of different bacterial diets have shown to significantly affect life span in invertebrate model systems such as *Caenorhabditis elegans* (*Zhao et al., 2013*), a functional test of the role of a complex GM community associated with young age as an intervention aimed at modulating the recipient's life span has not yet been carried out to date. By acutely exposing middle-age individuals to young fish GM content – after antibiotic treatment – we could prolong life span and retard the age-dependent decline in exploratory behavior, which we showed to decline during normal aging. Noteworthy, our results exclude that the effects of the interventions depend on repopulating the intestine with any GM community or that antibiotic treatment alone, delaying dysbiosis, was sufficient to explain the full extent of life span increase achieved via transfer of young GM.

Additionally, life span was not affected in young fish exposed to GM from old, young, and sham control fish after antibiotic treatment. These results are compatible with a scenario where the age-associated decline of immune function might be responsible for the progressively decreased capacity of the host to (i) maintain the healthy portion of the GM community and (ii) counteract the proliferation of potentially pathogenic gut bacteria. Since we observed that young fish receiving 16-week-old gut microbiota content did not live significantly shorter than young, control fish, it is plausible that dysbiosis, in the context of a young gut immune function, does not lead to increased mortality.

Fish treated with young GM after depletion of their own resident GM community not only maintained a more diverse microbial community at old age compared to wild-type, age-matched control fish, but their microbial community remained more similar to that of young fish. This raises the possibility that bacterial consortia associated with young fish can contribute to increased life span and enhanced individual health status. Based on functional metagenomic analysis, young fish and fish treated with young GM were enriched for bacteria associated with carbohydrate metabolism and DNA repair, both importantly associated with host metabolism, health and longevity.

Young fish, as well as fish treated with young GM, had a high number of bacterial taxa that frequently co-occurred with one another, de facto contributing to a young-associated bacterial network. On the other end, old wild-type controls (16wk), as well as old fish treated with same age GM (Omt), had smaller bacterial networks, possibly resulting from the higher inter-individual variation in GM composition associated with these groups. Remarkably, applying our analyses to a published mouse cohort (*Langille et al., 2014*), we extended this finding to mammals, confirming that networks built on GM OTU abundance are associated with host's chronological age. Our network analysis enabled us to identify a subset of highly frequent taxa associated with a young-like status and with prolonged life span in fish treated with young GM. These involved the genera *Exiguobacterium*, *Planococcus*, *Propionigenium* and *Psychrobacter*, which are key bacterial genera responsible for structuring a healthy GM community in TK. Interestingly, species belonging to each of these genera have been associated with energy metabolism and potential health benefits. Specifically, species of *Exiguobacterium* and *Propionigenium* are able to metabolize cellulose and ferment carbohydrates to produce short chain fatty acids, which are known anti-inflammatory mediators and can modulate the immune system. *Planococcus* species can hydrolyze gelatin to produce essential amino acids for use by the host and certain *Psychrobacter* species are capable of producing omega-fatty acids. Taken together, these key bacterial genera can produce metabolites capable of maintaining immune system health and having anti-inflammatory effects on the host, both of which have been associated with longevity.

While GM transfers significantly affected the GM composition of experimental fish, their overall gut transcriptional profile showed that old fish clustered together regardless of the treatment. This could be a consequence of the down-regulation of the transcriptional programs associated with growth in all 16-week-old fish groups. However, transcripts involved in defense against pathogens, extracellular matrix components and the Tor pathway, are dramatically different among experimental groups receiving young or same-age gut content, suggesting that these key aspects might ultimately be fundamental modulators of organismal life span and health. Generating a correlation matrix between bacterial abundance and gut transcripts, we could isolate host transcripts whose expression was significantly correlated with specific OTUs. In particular, bacterial genera associated with a healthier and longer life span, such as *Psychrobacter* and *Exiguobacterium*, had strong co-occurrence with host genes importantly associated with aging modulation.

The lack of a generalized transcriptional gut rejuvenation in long-lived fish treated with young GM, together with the fact that antibiotic-only treated fish live longer than control, untreated groups, suggest that a delayed onset of dysbiosis could benefit the host and explain, at least in part, the effects on survival. However, since fish receiving young GM live longer and have a young-like locomotor performance compared to the antibiotic-only treated group, it is likely that, independent from dysbiosis, young GM could directly benefit host physiology, possibly influencing metabolism and immune function. Therefore, young GM could possibly lead to beneficial effects on host survival and behavior performance both via delayed dysbiosis and direct favorable effects of subsets of young microbes. Future work will help shed light on what specific aspects of the aging process can be affected by resetting a young-like gut microbiota in aged individuals and whether this intervention can be more broadly applied to other organisms, including mammals.

Our results indicate that improving the ecological diversity of the GM in old individuals helps to restore health and prolongs life span. Our approach could provide a key to slowing aging and retarding the onset of age-associated diseases by specifically targeting the GM. Given its large bacterial taxonomic diversity and the shortest life span for a vertebrate species raised in captivity, the TK could become a key experimental species which will help to shed light on the functional connection between GM dynamics and aging in vertebrates.

## Materials and methods

### Wild fish samples

The wild fish samples were collected in 2015 during an expedition in the Gonarezhou National Park in(Permit No.: 23(1) (C) (II) 30/2015) and Mozambique (DPPM/069/710/11). Intestines were collected at each location and preserved in pure ethanol. Sampling locations coordinates are listed in *Figure 2—source data 1*.

### Fish husbandry and survival scoring

Fish (GRZ strain) used for microbiota analysis and scored for survival were individually housed from week 4 post-hatching in single 2.8L tanks connected to a water recirculation system receiving 12 hr of light and 12 hr of dark every day. Water temperature was set to 28°C and fish were fed blood worm larvae and brine shrimp nauplii twice a day during the week and once a day during the weekend. Dead fish were removed daily from the tanks, weighed and stored in 95% ethanol.

### DNA extraction, 16S rRNA gene amplification and sequencing

All dissected intestinal samples were collected at the same time prior to morning feeding and flash frozen in liquid nitrogen. For DNA isolation, frozen intestines were placed into autoclaved 2 ml screw caps tubes containing 1 ml of lysis buffer (80 mM EDTA, 200 mM Tris (pH 8.0) and 0.1M NaCl in PBS) and 0.4 g of a mixture of 0.1 mm zirconia/silica and 1.4 mm stainless steel beads (Biospec Products). Samples were bead-beaten for 3 min at 30 Hz (TissueLyzer II, Qiagen). Following the bead beating step, SDS (10% final concentration) and RNase A (PureLink, Invitrogen) were added and samples were incubated for 30 min at 55°C. DNA was then isolated using phenol:chloroform:isoamyl alcohol (Invitrogen) as per manufacturer's instructions with an additional chloroform step to remove excess phenol.

DNA was then used in one of two, two-step PCR methods designed to target the V3-V4 region of the 16 rRNA gene (*Klindworth et al., 2013*). For the first method, initial primers consisted of a 5'-to-3' primer-pad and linker (*Caporaso et al., 2012*) and the V3/V4 gene specific forward or reverse primer sequences. The second step PCR used primers complementary for the primer pad and linker followed by standard Illumina adaptors. The reverse primer for the second step also contained a 12 bp Golay barcode (*Caporaso et al., 2012*). For the second method, initial primers consisted of (5' to 3') Illumina overhang adaptor sequences and the V3/V4 gene specific primer sequences followed by a second step PCR using the Nextera XT Index kit (Illumina). For both amplification methods cycling conditions were the same and the first round of PCR was performed in triplicate with approximately equal amounts of DNA template (250 ng/reaction). PCR reactions were carried out with a two-minute denaturation step at 98°C, followed by 25 cycles of 98°C for 30 s, 55°C for 30 s and 72°C for 30 s. Triplicate reactions were then pooled and cleaned using the Wizard SV Gel and PCR Clean-up Kit (promega). The cycling conditions for the second step PCR were the same as the first step, except annealing was performed at 60°C with only eight cycles. Both PCR steps used KAPA HiFi Hotstart ReadyMix (KAPA Biosystems) and 1 μm of primers in 25 μl total volume. Second-step PCR products were run on a 1.2% agarose gel and DNA products between 500–700 base pairs were excised and cleaned up as in step 1. PCR products were quantified by Qubit (Life Technologies), diluted to 4 nm and combined in equal volumes. The combined amplicon libraries were then sequenced on the Illumina MiSeq, V3 reagents, 2 × 300 bp paired-end reads.

### Antibiotic treatment and fish microbiota transfer

Microbiota transfer experiments were based on those developed in zebrafish (*Pham et al., 2008*). Recipient fish were removed from main water recirculating system and housed in 9L tanks at a

density of 10 fish per tank. Recipient fish (6wk or 9.5wk) were treated overnight with a combination of Vancomycin (0.01 g/L), Metronidazole (0.5 g/L), Neomycin (0.5 g/L) and Ampicillin (0.5 g/L) to diminish the resident bacterial community. Following antibiotic treatment, recipient fish were washed twice for 10 min with autoclaved tank water. Concurrently, whole intestines were isolated from donor fish and placed into 10 cm petri dishes containing sterile PBS on ice. Intestines were then opened longitudinally, the intestinal contents scraped out and then further cut into 0.5 cm pieces to facilitate the release of bacteria. The collected intestinal contents were washed once in cold PBS and added to the fish tanks containing autoclaved tank water and recipient fish at a ratio of 1 donor fish intestine/2 recipient fish. Fish were incubated overnight with the donor fish intestinal contents before being returned to the main recirculating system and individually housed, where they were regularly fed according to standard husbandry.

## Microbial community analysis

Fastq files from paired end reads were joined, demultiplexed and subjected to quality filtering with QIIME 1.8 (Q $\geq$ 20) as previously described (*Caporaso et al., 2012*). For microbial community analysis, QIIME was used to identify OTUs by open-reference picking using UCLUST (97% similarity) and taxonomy was assigned with the Greengenes 13.8 database (*DeSantis et al., 2006*; *Edgar, 2010*). A minimum OTU count of 5 was used to minimize spurious OTUs. Representative OTU sequences were aligned with PYNAST (*Caporaso et al., 2010a*) and FastTree 2 (*Price et al., 2010*) was used to build a phylogenetic tree. For diversity analyses, OTU tables were rarefied to at least 5000 sequences, which allowed the majority of samples to be kept. QIIME and R were used to calculate alpha and beta diversity metrics and generate plots. To identify bacteria associated with specific groups, OTU tables were further filtered for presence in at least 25% of samples with a collective abundance of greater than 100 reads. Significant changes in relative OTU abundance were identified with linear discriminant analysis effect size (LEfSe [*Segata et al., 2011*]) and visualized using GraPhlAn (*Asnicar et al., 2015*). For metagenomics analysis, rarefied OTU tables were generated by closed reference picking and the PICRUSt tool was used to normalize by 16S copy number and predict the metagenome content of samples from 16S rRNA profiles. KEGG pathway functions were then categorized at level 2 or 3 and LEfSe was used to identify significant changes among classes. The accuracy of PICRUSt predictions was determined by nearest sequenced taxon index (NSTI).

## Bacterial diversity among model organisms

To compare the overall bacterial diversity among different species, sequences were downloaded from the European Nucleotide Archive (ENA), the NCBI Short Read Archive (SRA) or MG-RAST and subjected to closed reference OTU picking. Taxa summary and alpha diversity measures were computed using QIIME 1.8. Samples were chosen which used Illumina sequencing of the V4 region of the 16 rRNA gene. Accession numbers. Human: ERR561021UK, ERR560915UK, ERR560902UK, ERR560855UK, mgm4489670, mgm4489628, mgm4489516, mgm4489454, mgm4538473, mgm4538468, mgm4538264, mgm4538259, mgm4538238. Mouse: SRR1820108, SRR1820074, SRR1820073, SRR1820072, SRR1820071, ERR706143, ERR706142, ERR706141. Zebrafish: SRR1581750, SRR1581753, SRR1581759, SRR1581763, SRR1581766, SRR1581889, SRR1581890, SRR1581891. Drosophila: SRR989472, SRR989473, SRR989474, SRR989469, SRR989467, SRR952981.

## Bacterial load quantification

To determine bacterial load, DNA was isolated from freshly collected faecal pellets as described above. Real-time quantitative PCR was performed with DyNAmo Color Flash SYBR green master mix (Thermo Scientific) and run with the BioRad CFX384 Real-Time System. Samples were normalized to amount of input faecal material and graphed as 1/Ct value. Primers targeting the 16S gene were described previously (*Caporaso et al., 2010b*). For: 5' TCCTACGGGAGGCAGCAGT 3' and Rev: 5' GGACTACCAGGGTATCTAATCCTGTT 3'.

## Quantitative analysis of spontaneous locomotor activity

To measure total swimming distance, fish were placed in rectangular (16 $\times$ 90 cm) tanks filled with 8 liters of water and allowed to acclimate for 30 min. The room and water temperature and water

quality were kept as similar to the fish's' normal environment as possible. The fish were then filmed for 20 min with an overhead mounted camera and data were analyzed using EthoVision XT11 (Noldus Information Technologies).

## Co-occurrence networks

The OTU table was filtered by a minimum total count of 5509 per individual, based on the rarefaction value (see Microbial community analysis). Subsequently, the table was divided into subsets corresponding to the treatment groups. Normalization was achieved through dividing the OTU sample count by the total sample count, scaled to 1000.

Co-occurrence networks were produced by applying the SparCC program (*Friedman and Alm, 2012*), a network inference tool specifically developed for analysis of correlations in compositional data, such as 16S microbiome analyses. To avoid unreliable correlations for very rare OTUs, the OTU tables were filtered for bacteria that were present in at least 25% of samples of each group.

The SparCC pipeline was used to first calculate OTU-OTU correlations averaged over 20 iterations (*Friedman and Alm, 2012*). We then tested the significance of these correlations by computing pseudo p values against 1000 bootstrap simulations, applying the same parameters. All OTU-OTU correlations with a p value lower than 0.05 were considered significant and were included into network preparation. Nodes without any edges after filtering by significant p values were removed from the network. To analyze the network properties we used the igraph package in R, version 1.0.1 (*Csardi and Nepusz, 2006*). For each group we generated an undirected network, weighted by correlation magnitude. For biological interpretation we only focused on positive correlations. To identify network size and composition, we calculated the negative cumulative degree distribution and the percentage of genus present in the network belonging to different phyla. The main clusters of each network were identified with two different clustering algorithms of the igraph package (*Csardi and Nepusz, 2006*). The first algorithm 'k-core' clusters based on the degree, with each member of the maximal subgraph has at least a degree count of k. The second algorithm 'infomap community' searches for community structures that minimize in the length of a random walker trajectory. We used the overlap of the largest clusters within the two cluster algorithms to identify the main clusters in each group. The members of clusters from young fish (6wk) and old fish (16wk) were used to identify young-like or old-like clusters in the main clusters of the transfer groups (Ymt, Omt and Abx).

To compare our findings to a different model organism, we used a published data set (*Langille et al., 2014*). We scaled the relative frequencies from young, middle-age and old mice to 1000 and excluded bacteria not present in at least 25% of samples. Additionally, we excluded sample Y7.August14, because we included the same sample from 15th of August (Y7.August15). Following the same approach to the previous analysis, we calculated the negative cumulative degree distribution.

## RNA extraction

Intestines from four young (6-weeks-old), four old (16-weeks-old), four old fish receiving 6-week-old donor fish gut content (Ymt) and four old fish receiving 9.5-week-old fish gut content (Omt) were dissected. Trizol (15596018, Thermo Fisher Scientific, USA) was used to isolate total RNA, following manufacturer's instructions. Residual genomic DNA was removed by DNaseI treatment (AM2222, Thermo Fisher Scientific, USA) and 1.5 μg of total RNA was used as a template for reverse transcription (11754050, Thermo Fisher Scientific, USA).

## RNA sequencing

We sequenced four individuals of groups 6wk, 16wk, Ymt and Omt. Each sample sequenced with 100 bp paired end with 30 million reads. All samples were mapped to the African Turquoise Killifish Genome (*Valenzano et al., 2015*) using STAR version 2.4.1 c (*Dobin et al., 2013*) with the default parameters, except for splice junction and chimeric junction parameters set to 15 base-pairs, as well as allowing for only five mismatches over the whole fragment and removing all reads that map to more than two locations.

## Differential gene expression

The differential gene expression was performed using featureCounts (*Liao et al., 2014*) and the edgeR package (*Robinson et al., 2010*) version 3.16.5. featureCounts was used to generate raw counts using the mapped reads and the annotation file from *Valenzano et al. (2015)*. The edgeR package was used to identify differentially expressed genes between any of the groups resulting in six pairwise comparisons. Additionally, we identified young-like genes that are differentially expressed between (i) 6wk and both, 16wk and Omt; (ii) Ymt and 16wk; but are not differentially expressed between (iii) Omt and 16wk; (iv) Ymt and 6wk. All edgeR runs were performed using TMM normalization, followed by tagwise dispersion and an exact test (*Robinson et al., 2010*). We considered only genes that had a counts-per-million value above 0.7 in at least 50% of compared samples. The resulting p values were multiple testing corrected with the Benjamini-Hochberg method. Only genes with a FDR smaller or equal than 0.1 were considered as differentially expressed. Gene enrichment analyses were performed using the GOrilla software (*Eden et al., 2009*).

## Functional analysis of target genes

To gain insights into the function of genes differing between young and old fish as well as genes differing between fish of long or normal lifespan we used the GOrilla software (*Eden et al., 2009*), which performs enrichment tests. We used the zebrafish database for annotation and the available reciprocal blast information (*Valenzano et al., 2015*). If top differentially expressed genes were not found to have a zebrafish ortholog in this resource, we used blastp (*Altschul et al., 1990*) to identify the corresponding zebrafish ortholog (*Figure 7—source data 1*). First, all genes were mapped to a GO-Term. The statistical enrichment test uses a hypergeometric distribution to test for enrichment in the ratios of target genes associated with a GO-Term, and the ratio of background genes associated with the same GO-Term to identify pathways or GO-Terms that have a significantly larger amount of target genes than expected. Here, we used all genes that were found to be expressed in the intestine of the experimental groups as background. The GOrilla results were visualized with the ggplot2 package in R.

## Correlation between individual OTU and gut transcripts

Individual fish OTU tables and log2-transformed transcripts were used as a source to identify significant correlations using Pearson's correlation.

The python code used to perform this analysis was saved on github (*Valenzano, 2017*). A copy is archived at https://github.com/elifesciences-publications/publications.

## Acknowledgements

We thank the Cologne Center for Genomics (CCG) and Paul Higgins for technical support with the sequencing, all the members of the Valenzano lab for their scientific input to the project. We are thankful to Adam Antebi and Linda Partridge for discussions, Anne Brunet, Anthony Zannas, Jenny Regan and Matthias Platzer for critically reading the manuscript. We thank the Research Council of Zimbabwe, Zimbabwe Parks and Wildlife Management Authoriry, Patience Gandiwa, Daphine Madhlamoto, Itamar Harel, Radim Blažek, Matej Polačik, Tamuka Nhiwatiwa, Hugo and Elsabe van der Westhuizen, Evious Mpofu, Power Mupunga and all the rangers of the Gonarezhou National Park for their support with the fieldwork. All animal experiments of this study have been examined and approved by the competent authority (Landesamt für Natur, Umwelt und Verbraucherschutz Nordrhein-Westfalen; AZ: 84–02.04.2015.A377).

## Additional information

### Funding

| Funder | Grant reference number | Author |
|---|---|---|
| Czech Science Foundation | 16-00291S | Martin Reichard |
| Max Planck Institute for Biol- | Open-access funding | Dario Riccardo Valenzano |

ogy of Ageing

The funders had no role in study design, data collection and interpretation, or the decision to submit the work for publication.

## Author contributions

PS, Conceptualization, Data curation, Formal analysis, Investigation, Writing—original draft; DW, Data curation, Formal analysis, Visualization, Methodology, Writing—review and editing, wild specimens acquisition; MP, Data curation, Formal analysis, Investigation, Visualization, Writing—review and editing; FM, Data curation, Formal analysis; EG, Conceptualization, Writing—review and editing; MR, Resources, Investigation, Writing—review and editing, wild specimens acquisition; DRV, Conceptualization, Resources, Data curation, Formal analysis, Supervision, Funding acquisition, Investigation, Visualization, Methodology, Writing—original draft, Project administration, Writing—review and editing, Field work coordination, wild specimens acquisition

## Author ORCIDs

Martin Reichard, http://orcid.org/0000-0002-9306-0074
Dario Riccardo Valenzano, http://orcid.org/0000-0002-8761-8289

## Ethics

Animal experimentation: Landesamt für Natur, Umwelt und Verbraucherschutz Nordrhein-Westfalen; AZ: 84-02.04.2015.A377

# Additional files

## Major datasets

The following datasets were generated:

| Author(s) | Year | Dataset title | Dataset URL | Database, license, and accessibility information |
|---|---|---|---|---|
| Smith P, Willemsen D, Popkes M, Metge F, Gandiwa E, Reichard M, Valenzano DR | 2017 | Turquoise killifish (Nothobranchius furzeri) gut microbiota | https://www.ncbi.nlm.nih.gov/bioproject/PRJNA379271 | Publicly available at the NCBI BioProject database (accession no: PRJNA379271) |
| Smith P, Willemsen D, Popkes M, Metge F, Gandiwa E, Reichard M, Valenzano DR | 2017 | RNAseq of turquiose killifish (Nothobranchius furzeri) in ageing and after gut microbiota transfer | https://www.ncbi.nlm.nih.gov/bioproject/PRJNA379208 | Publicly available at the NCBI BioProject database (accession no: PRJNA379208) |

The following previously published dataset was used:

| Author(s) | Year | Dataset title | Dataset URL | Database, license, and accessibility information |
|---|---|---|---|---|
| Beiko R | 2014 | Mice Frailty (mgp3907) | http://metagenomics.anl.gov/linkin.cgi?project=mgp3907 | Mice Frailty (mgp3907) |

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
