## [Decision Letter]

Thank you for submitting your article "Regulation of Life Span by the Gut Microbiota in The Short-Lived African Turquoise Killifish" for consideration by *eLife*. Your article has been reviewed by three peer reviewers, and the evaluation has been overseen by a Reviewing Editor and Detlef Weigel as the Senior Editor. The following individuals involved in review of your submission have agreed to reveal their identity: David Walker (Reviewer #1).

The reviewers have discussed the reviews with one another and the Reviewing Editor has drafted this decision to help you prepare a revised submission.

Smith et al. present evidence for an influence of the microbiota on longevity in the African Turquoise Killifish. The authors characterize age-related changes in the microbiota and test whether transfer of microbiota from young fish can impart longevity and modulate fitness traits as well as intestinal transcriptional profiles in aging fish. Based on these studies, the authors conclude that the fish gut microbiota becomes less diverse with age, that microbiota transfer can extend lifespan and delay the onset of age-related motility defects, that the composition of the microbiota can be changed towards a younger phenotype in old fish by the transfer of microbes from young fish, but that the transcription profile of aging intestines is not significantly affected by the transfer of a young microbiota. The authors point out a few genes that are differentially expressed in response to young microbiota transfer and highlight changes in microbiota networks that suggest restoration of higher connectivity by the transfer of a young microbiota.

While all three reviewers find enthusiasm for this work, we believe that clarity in the interpretation of the results and experiments needs to be performed before publication. Below we have consolidated the reviews to outline the major issues that need to be addressed.

It is stated in the manuscript that "bacterial composition changes with age but that there was no change in bacterial abundance.". Are the authors claiming that there was no change in overall bacterial load in aged fish compared to young fish? The data showing this is not clear and not consistent with fly and mouse studies.

Have replicate lifespan experiments been performed?

Can the authors please include a detailed discussion of how the young microbiota transfer provides a beneficial effect to the fish or merely delays the onset of dysbiosis. Both are equally possible, but not discussed equally.

While we agree that the lifespan effect is robust, can the authors speculate about ageing? Is this delayed and which "Hallmarks" could be quickly assessed in the fish. If none, please bring this point into the Discussion.

The authors have chosen 9.5-week old fish as a recipient for gut microbiota transfers from young fish in Figure 4. It would be nice to have the analysis of gut microbiota of 9.5-week old fish given that it may reflect either young or old gut states. Does it differ significantly from microbiota of 6-week-old or 16-week-old fish?

In the present manuscript, Figure 7 describes a differential transcription profile of genes associated with TOR-pathway and cell adhesion and extracellular matrix composition when comparing Ymt and 16-week-old fish. Are these profiles also significantly different when comparing Ymt with the other antibiotic treated conditions (Omt and Abx)?

---

## [Author Response]

[…] While all three reviewers find enthusiasm for this work, we believe that clarity in the interpretation of the results and experiments needs to be performed before publication. Below we have consolidated the reviews to outline the major issues that need to be addressed.

It is stated in the manuscript that "bacterial composition changes with age but that there was no change in bacterial abundance.". Are the authors claiming that there was no change in overall bacterial load in aged fish compared to young fish? The data showing this is not clear and not consistent with fly and mouse studies.

This is a very important point and we thank the reviewers and the Editor to point it out. Indeed, although we find that old fish, compared to young fish, have significantly lower bacterial richness (alpha diversity, Figure 3), overall stool bacterial load was unchanged between young and old fish (Figure 3—figure supplement 1). Since the measure of bacterial load was obtained from stool bacterial content, we believe it does not necessarily conflict with the current literature on fly and mouse studies. We have now better clarified this point in the Results section; toning down the interpretation that bacterial abundance was globally unchanged in the two groups.

Have replicate lifespan experiments been performed?

We thank the reviewers for requesting to better clarify this key point in the manuscript. Indeed, we replicated the survival experiments four times. This is indicated in Figure 4—figure supplement 1 and all the data relative to the survival experiments are available in Supplementary file 3. We now better clarify this in the Results section.

Can the authors please include a detailed discussion of how the young microbiota transfer provides a beneficial effect to the fish or merely delays the onset of dysbiosis. Both are equally possible, but not discussed equally.

We agree that the results presented in our work require a thorough discussion of how the young gut microbiota transfer could benefit the host. To better clarify either scenario laid out, we now elaborate more extensively in Discussion on how microbiota transfers could prolong lifespan by retarding the onset of dysbiosis and/or by slowing down host ageing. Given our results, we believe that although delayed dysbiosis might be responsible – at least in part – for the beneficial effects of microbiota transfer (antibiotic-treated fish live longer than control fish), the fact that fish treated with young gut microbiota outlive fish treated with antibiotics-only suggests that young gut microbes might indeed play a directly beneficial effect on host physiology.

While we agree that the lifespan effect is robust, can the authors speculate about ageing? Is this delayed and which "Hallmarks" could be quickly assessed in the fish. If none, please bring this point into the Discussion.

Whether the effect of transferring young gut microbiota to middle-age fish has only effects on survival but none on aging is an outstanding question. In our work, we mainly focused on survival. However, our behavioral assay in fish also demonstrates that overall decline of physical fitness, measured by spontaneous locomotor activity, is also delayed by young gut microbiota transfer. Importantly, at 16 weeks of age, fish treated with young gut microbes have significantly higher activity levels compared to all other groups at the same age, including the antibiotic-only treated group (Abx). To point this out, we now updated Figure 4, which previously did not report this result. Our choice to measure levels of locomotor activity was based on the fact that behavioral assays are highly integrated measures of overall health, and therefore provide a good indication of overall aging. Additionally, we have previously shown that exploratory behavior in the turquoise killifish significantly declines with age. We interpret our result as a strong indicator that treating fish with young gut microbes improves overall health status and delays (at least) motor aging. However, we agree with the reviewers that we do not provide additional measures of fish aging and whether gut microbiota transfer affects them. These aspects urgently need to be addressed in future work.

The authors have chosen 9.5-week old fish as a recipient for gut microbiota transfers from young fish in Figure 4. It would be nice to have the analysis of gut microbiota of 9.5-week old fish given that it may reflect either young or old gut states. Does it differ significantly from microbiota of 6-week-old or 16-week-old fish?

We thank the reviewers and the editor for this question. We acknowledge that in the text we did not explain that on Figure 5—figure supplement 1, we provide four plots where we compare alpha and beta diversity between all the experimental groups, including 10-week-old wild type (10wk), i.e. the same age at which we perform the transfer. The results show that the 10wk group has a somewhat intermediate behavior between young (6wk) and old (16wk) fish. While 10wk fish have an alpha diversity similar to young wild type fish (Figure 5—figure supplement 1), their beta diversity is more similar to that of older fish (Figure 5—figure supplement 1).

To further address this point, we performed new clustering analysis of wild type fish at 6, 10 and 16 weeks of age (Figure 8), which shows that 10wk fish do not preferentially cluster with either young (6wk) or old (16wk) fish. Additionally, we measured beta diversity distances (Bray-Curtis) between all the wild-type groups, and found that the 10wk group has similar beta diversity as the 6wk and 16wk group (Figure 9), i.e. 10wk fish are not significantly closer to 6wk compared to 16wk.

Author response image 1.Cluster dendrogram between 6wk-10wk-16wk wild-type killifish gut microbiota.**DOI:**
http://dx.doi.org/10.7554/eLife.27014.029

Author response image 2.Bray-Curtis beta diversity among wild-type age-groups.**DOI:**
http://dx.doi.org/10.7554/eLife.27014.030

Overall, these results indicate that 10wk fish are distinct from 6wk and 16wk fish, possibly showing an intermediate behavior. We added a sentence in the discussion to clarify that, based on alpha and beta diversity analyses, we can support that 10wk fish do not belong to either the young (6wk) nor to the old (16wk) groups. We thank again the reviewers and the editor for this comment, as we believe that clarifying this result strengthens our manuscript.

In the present manuscript, Figure 7 describes a differential transcription profile of genes associated with TOR-pathway and cell adhesion and extracellular matrix composition when comparing Ymt and 16-week-old fish. Are these profiles also significantly different when comparing Ymt with the other antibiotic treated conditions (Omt and Abx)?

First, we would like to report that we have refined the expression data analysis, because the initially used read-count table from the cufflink-suite was generated with cuffnorm, which uses an internal normalization procedure that might interfere with the normalization assumptions performed downstream in edgeR. Therefore, we generated a new raw read-count table with featureCounts and subsequently used this table for the downstream analysis with edgeR. This resulted in updated Figure 7. The new GO analysis show the same results as before, i.e. defence response to bacteria is increased in Ymt and increased expression of genes associated with hyaluronic acid and mucopolysaccharides metabolism is increased in Omt (Figure 7). Then, we re-analysed gene expression based on significantly differential expression between 6wk and Omt-16wk, and between Ymt and 16wk. We did not perform RNA-sequencing on Abx fish. The list of genes that were differentially expressed, after a multiple-hypothesis testing correction (BH FDR = 0.1) still includes DEPTOR and DSCAM, confirming our previous results (Figure 7). Additionally, both DEPTOR and DSCAM were also differentially expressed between Ymt and Omt (p values = 0.001 and 0.022, respectively). However, these differences were not significant after multiple hypothesis correction with an FDR of 0.1, which is the reason why we do not show significance bars between Ymt and Omt in Figure 7. Among the differentially expressed genes, after multiple hypothesis correction, using an FDR = 0.1, both OLFML2A and PAPLNB were excluded from the new analysis. These two genes were excluded as they did not meet the previously described criteria, i.e. OLFML2A was not differentially expressed between Ymt and 16wk, and PAPLNB was not differentially expressed between 6wk and Omt (Figure 10).

Author response image 3.Differential gene expression in OLFM2A(1of2) and PAPLNB.**DOI:**
http://dx.doi.org/10.7554/eLife.27014.031

Here we provide the statistics relative to the DEG analysis in OLFML2A and PAPLNB (Table 1).

Author response table 1.Differential gene expression values for OLFML2A and PAPLNB.**DOI:**
http://dx.doi.org/10.7554/eLife.27014.032Comparison\geneOLFML2A – FDROLFML2A – p valuePAPLNB – FDRPAPLNB – p value6wk-16wk0.0000.0000.0000.0006wk-Omt0.0260.0020.1670.0276wk-Ymt0.0530.0080.7750.544Ymt-16wk0.2160.0040.0510.000Ymt-Omt1.0000.6501.0000.110Omt-16wk0.2560.0140.3670.032

Overall, we re-analyzed the RNASeq data using edgeR and could confirm that the same GO terms as before were significantly different between Ymt and Omt. Additionally, DSCAM and DEPTOR were differentially expressed between 6wk and Omt/16wk, and between Ymt and 16wk, but were not differently expressed between Ymt and Omt after multiple hypothesis correction. Finally, the new set of differentially expressed genes excluded OLFML2A and PAPLNB from the list previously reported. These results were updated in Figure 7, Results and in the Methods section.